behaviour

collective navigation, fishway, anthropogenic barriers, survival modelling, migration

**Authors for correspondence:**
Connie Okasaki
e-mail: cokasaki@uw.edu
Andrew M. Berdahl
e-mail: berdahl@uw.edu

# Collective navigation can facilitate passage through human-made barriers by homeward migrating Pacific salmon

Connie Okasaki[1,2], Matthew L. Keefer[3], Peter A. H. Westley[4] and Andrew M. Berdahl[1,2]

[1]Quantitative Ecology and Resource Management Program, and [2]School of Aquatic & Fishery Sciences, University of Washington, Seattle, WA 98195, USA
[3]Department of Fish and Wildlife Sciences, College of Natural Resources, University of Idaho, Moscow, ID 83844-1136, USA
[4]Department of Fisheries, University of Alaska Fairbanks, Fairbanks, AK 99775, USA

 CO, 0000-0003-3155-966X; MLK, 0000-0002-4264-2576; PAHW, 0000-0003-4190-7314; AMB, 0000-0002-5057-0103

The mass migration of animals is one of the great wonders of the natural world. Although there are multiple benefits for individuals migrating in groups, an increasingly recognized benefit is collective navigation, whereby social interactions improve animals' ability to find their way. Despite substantial evidence from theory and laboratory-based experiments, empirical evidence of collective navigation in nature remains sparse. Here we used a unique large-scale radiotelemetry dataset to analyse the movements of adult Pacific salmon (*Oncorhynchus* sp.) in the Columbia River Basin, USA. These salmon face substantial migratory challenges approaching, entering and transiting fishways at multiple large-scale hydroelectric mainstem dams. We assess the potential role of collective navigation in overcoming these challenges and show that Chinook salmon (*O. tshawytscha*), but not sockeye salmon (*O. nerka*) locate fishways faster and pass in fewer attempts at higher densities, consistent with collective navigation. The magnitude of the density effects were comparable to major established drivers such as water temperature, and model simulations predicted that major fluctuations in population density can have substantial impacts on key quantities including mean passage time and fraction of fish with very long passage times. The magnitude of these effects indicates the importance of incorporating conspecific density and social dynamics into models of the migration process. Density effects on both ability to locate fishways and number of passage attempts have the potential to enrich our understanding of migratory energetics and success of migrating anadromous salmonids. More broadly, our work reveals a potential role of collective navigation, in at least one species, to mitigate the effects of anthropogenic barriers to animals on the move.

## 1. Introduction

Long-distance migration is an iconic and threatened behaviour [1]. Migratory species navigate with incredible precision to and from highly spatially restricted locations [2]. To solve these challenging navigational problems, species use mechanisms ranging from an innate sun compass [3] and magnetic maps [4] to learned olfactory cues [5,6]. However, long-distance migrations are threatened by human influence [7]. Fences, highways and other developments block terrestrial migration pathways [8], light pollution interferes with aerial migrations [9] and dams and de-watering impede passage both upstream and downstream (e.g. Norrgård *et al.* [10]). In light of these impacts, it is essential to understand the mechanisms of navigation in order to predict and mitigate human impacts on migratory populations.

Many species migrate in groups, and this is thought to aid navigation [11]. Theory suggests that such *collective navigation* may be the result of a number of mechanisms. For example, groups can increase accuracy by averaging over error-prone individual directional estimates—known as the 'many wrongs' principle [12]. Even without individual directional estimates, accurate directional responses can emerge through social interactions—known as 'emergent sensing' [13]. We direct interested readers to box 1 of Berdahl *et al.* [11] for an overview of these and other mechanisms including leadership [14], and social and collective learning [15,16]. A growing body of empirical literature lends compelling support to these hypothesized mechanisms, but evidence from wild populations is rare [11].

One iconic example of a migratory species is salmon, which home in large numbers back to their natal spawning grounds. Salmon navigation is not fully understood but is known to include an inherited magnetic map [4] and olfactory recognition of natal water [5]. Berdahl *et al.* [17] hypothesized that salmon use collective navigation, based on seven independent studies reporting positive associations between homing accuracy and run size. However, like many migratory species, salmon face an anthropogenic barrier: dams. For example, the impassable Swan Falls Dam on the Snake River in Idaho rendered approximately 25% of mainstem riverine habitat inaccessible to Snake River Chinook salmon (*Oncorhynchus tshawytscha*). A study of fall-run Chinook spawning habitat on the Columbia River found that between impassable dams and altered flow regimes, less than 20% of historical spawning habitat for fall-run Chinook remained available [18]. Many dams have fishways, which allow salmon to pass by these barriers. However, locating a fishway entrance is non-trivial, since salmon use rheotaxis to move upstream, and the main source of flow at most dams is the spillway or turbines. Adult salmon migrations are often delayed in dam tailraces, and this delay may bear important costs, including increased exposure to predation (e.g. Keefer *et al.* [19]) and increased energy output which can lead to greater mortality (e.g. Burnett *et al.* [20]). Thus, if collective navigation eases the dam passage process, it may have an important effect on salmon survival and reproductive success, and therefore important implications for population conservation.

Here, we use radiotelemetry data on adult Chinook and sockeye salmon (*Oncorhynchus nerka*) navigating upstream past dams on the Columbia River, combined with daily fish counts at the dams, to evaluate the hypothesis that collective navigation helps salmon overcome the navigational challenges posed by fishways. We find strong evidence that Chinook salmon find and commit to fishways more rapidly on higher-density days. Evidence for density effects in sockeye salmon, and for Chinook salmon navigating within fishways, was weak. Although some prior evidence exists for negative density effects in similar contexts (e.g. Goerig & Castro-Santos [21]), we did not find any prior evidence of negative density effects in either sockeye or Chinook salmon.

## 2. System and methods

### (a) Study system

The Columbia River drains greater than 600 000 km² of seven western US states and two Canadian provinces and historically supported some of the most abundant Pacific salmon and steelhead runs in the world [22]. The basin has been transformed by hydroelectric development, with 14 large dams on the main stem Columbia River and 20 dams on the main stem Snake River, the Columbia's largest tributary by area. The dams, along with overharvest, habitat loss and artificial propagation, contributed to steep declines in Columbia River salmon populations [23] and subsequent threatened or endangered status under the US Endangered Species Act [24].

Upstream-migrating adult salmonids can pass many of the Columbia basin dams via pool-and-weir fishways [25] that rise $\approx 17$–$56$ m per dam. To navigate past the dams, adults first pass through turbulent, high-velocity tailraces that are several kilometres long and greater than 1 km wide. Fish must then locate low-volume fishway openings sited near powerhouses or adjacent to spillways, move through a series of collection channels and junction pools, ascend a fish ladder, and then exit into the upstream reservoir (figure 1). The spatial scale and hydraulic complexity of dam tailraces and fishways present several navigational and physiological challenges. The combination of searching for passage routes and fishway exit and re-entry behaviours, for example, is energetically demanding, particularly when fish make multiple passage attempts [26,27]. Typical upstream migration rates for Chinook and sockeye salmon in undammed sections of the Columbia basin range from $\approx 18.5$–$52.7$ km/day [28]. By contrast, adult salmonids take $\approx 1$–$3$ days to pass each main stem dam along their Columbia River migration route [29]. These tailrace and dam reaches range in length from 0.5 to 3.2 km, such that, on average, the fish are travelling 0.17–3.2 km/day during dam passage (a $\approx 10$–$100$-fold reduction in up-stream passage speed), potentially delaying timely arrival at spawning sites.

### (b) Data collection

Data used in this study were from salmon collected and radio-tagged at Bonneville Dam in 2013 and 2014 using previously described methods [29,30]. Bonneville Dam is at Columbia River kilometre (rkm) 235.1 and is the first dam returning adult salmon encounter during their upstream migration. Telemetered fish were monitored at multiple dams and in tributaries, but analyses focused on data collected as fish entered tailraces and passed through fishways at The Dalles Dam (TD; rkm 308.1) and John Day Dam (JD; rkm 346.9), the second and third dams from the Pacific. Monitoring arrays at the two projects included aerial nine-element Yagi antennas sited on tailrace shorelines 1.8–3.2 rkm downstream from the dams and underwater coaxial cable antennas at fishway openings, inside fishway collection channels and junction pools, and in fish ladders [29,31]. Detection ranges for the aerial antennas ranged from hundreds of metres to greater than 1 km, depending on fish depth [32]. Ranges for the underwater antennas were 5–15 m. The raw telemetry data from all antennas were assembled, filtered and coded using established methods [29,30]; files of coded records were then used to identify when salmon entered and exited tailraces and fishways.

We obtained daily counts of salmon from the Columbia Basin Research Data Access in Real Time (DART) database [33] and environmental data including water temperature and spillway discharge from the US Geological Survey's (USGS) National Water Inventory System (NWIS) database (https://waterdata.usgs.gov/nwis). This synthesis of several different sources of data was crucial for our analysis. Daily counts are collected by human observers and are able to capture a large fraction of passing fish. These data are therefore

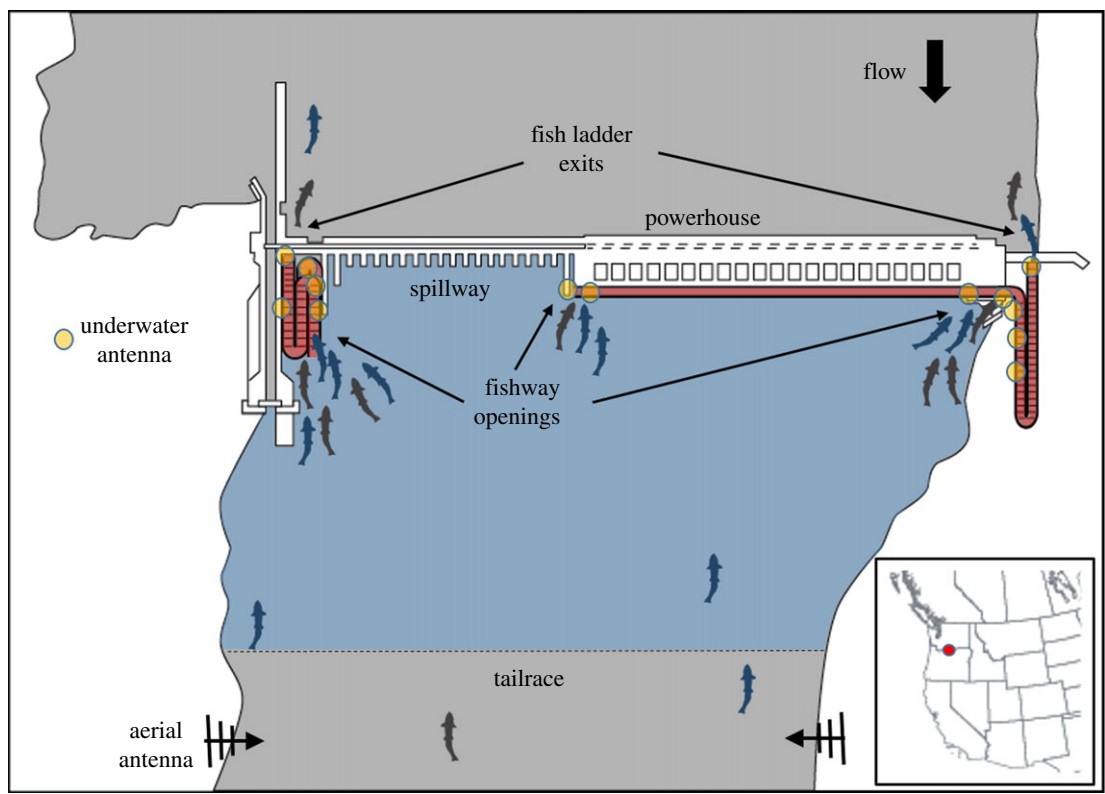

**Figure 1.** Schematic of John Day Dam. Layout of The Dalles Dam is qualitatively similar. The John Day facility is 2327-m long and 56-m high and has two adult fishways: one on each shoreline. The tailrace antennas were 1.8 km downstream from the dam (3.2 km at The Dalles Dam). Multiple underwater antennas were used to monitor fish passage into and through the fishways. Components of this schematic are not to scale. We modelled three processes: the 'finding' time from when a fish enters a tailrace (blue) to when it first enters a fishway (red); the 'fishway' time from when a fish last enters a fishway to when it exits into the upstream reservoir; and the 'commit' probability that a fish passes all the way through a fishway on its first attempt. (Online version in colour.)

suited to measuring density, but not to modelling individual passage rates. On the other hand, telemetry arrays are more expensive and technical, capturing only a small fraction of fish, but allow much more detailed observations of those fish which are tagged. These data allowed us to model passage rates for individual fish. Although the synthesis of these datasets was of great benefit, it also introduced several challenges, discussed in the following section. Most notably, counts were *not* a direct measurement of fish density.

## (c) Modeling approach

We split the process of passing a dam into a sequence of three distinct stages: (i) the 'finding' process which starts at a fish's first detection in a tailrace and continues until its first detection at a fishway opening; (ii) the 'commiting' process which models the probability that a fish actually passes all the way through a fishway to the upstream exit on its first attempt, rather than exiting back into the tailrace; and (iii) the 'fishway' process which starts when a fish enters a fishway and commits (i.e. the last fishway entrance) and continues until the fish exits from the top of the ladder.

We modelled the 'finding' and 'fishway' processes using a time-to-event analysis (also known as 'survival' analysis in the medical literature) framework (for a general introduction, see Kleinbaum & Klein [34]). We modelled the 'committing' process using a logistic regression. A detailed technical description of our models and diagnostics can be found in the electronic supplementary materials [35–40].

A time-to-event dataset consists of a series of observations through time of an individual along with a set of covariates for

that individual. We were interested in two events—that of entering a fishway, and that of exiting a fishway into the upstream reservoir—and we wished to understand how long it took for these events to occur. For each event, we must also define a starting time: respectively, the entrance to the tailrace and the last entrance to the fishway. In other words, our first model answers the question: 'how quickly does a fish find and enter a fishway after entering a tailrace?' Our second model answers the question: 'how quickly does a fish navigate and exit a fishway after entering it and committing to passage?' More complex 'multi-state' models tie events together into a sequence [41], but to simplify our analysis we used more tractable single-process models.

We used a common time-to-event model known as the proportional hazards (PH) model. A hazards model assumes that the time-to-event process is essentially an exponential decay, with an event rate $\lambda(t|X)$ (the 'hazard function') which changes depending on both how long it has been since the fish entered the system, $t$, and with a set of covariates $X$. The PH model assumes that this rate depends on covariates according to the equation

$$\lambda(t|X) = \lambda_0(t) \exp(X\beta).$$

The function $\lambda_0(t)$ is known as the baseline hazard function. In our case, covariates such as temperature, spillway discharge and fish density were time-varying and the rate at which fish passed through the dam was assumed to also vary in time proportionately.

Our focus was on the effect of density, which we quantified using daily counts of individuals obtained from the DART database [33]. This use of counts introduced two major problems. First, our count covariate was a product of density and hazard:

**Table 1.** Results for the 13 models considered after model splitting. Final model was not fit due to small sample size. All *p*-values calculated using parametric bootstraps from the AICc-selected null model. All 95% CIs calculated using parametric bootstrap from the full model (selected null model plus a density effect). Sample size reflects the number of radio-tagged fish. Note that the *p*-values were not Bonferroni corrected; see electronic supplementary material for more details.

| species | process | dam | subset | *p*-value | 95% CI | sample size |
|---|---|---|---|---|---|---|
| Chinook | find | JD | · | $<5 \times 10^{-04}$ | (0.36, 0.75) | 804 |
| | | TD | · | $<5 \times 10^{-04}$ | (0.16, 0.49) | 751 |
| | commit | · | · | $<5 \times 10^{-04}$ | (0.29, 1.2) | 930 |
| | fishway | JD | spring run | >0.1 | (−0.088, 0.49 ) | 439 |
| | | | summer run | >0.1 | (−0.59, 1.1) | 345 |
| | | TD | spring run | >0.1 | (−0.17, 0.2) | 415 |
| | | | summer run | 0.061 | (−0.054, 1.2) | 332 |
| sockeye | find | JD | · | >0.1 | (−0.032, 0.04 ) | 609 |
| | | TD | · | >0.1 | (−0.023, 0.042 ) | 616 |
| | commit | · | · | >0.1 | (−0.053, 0.075) | 678 |
| | fishway | JD | · | >0.1 | (−0.047, 0.028) | 605 |
| | | TD | east fishway | 0.037 | (0.00067, 0.039) | 553 |
| | | | north fishway | · | · | 59 |

when fish were passing through the fishway more quickly, a higher proportion of them would be counted over the course of a day. This introduced reverse causality: a high rate (the response) caused our counts (a predictor) to be higher. Moreover, since hazard varied substantially over the course of a day, the relationship between counts and density was inconsistent. Second, the fish observation windows are placed near the middle of fish ladders. Thus there was an unknown amount of time between when a fish triggered a telemetry reading and when it was counted by the observers. A third concern, not related to our use of counts, was that density is related to hazard through time: when fish are passing through a fishway more quickly, fewer of them remain behind to contribute to density. In other words, high passage rates deplete density over time. This introduced another source of reverse causality.

These issues posed a substantial challenge. The first and last were particularly worrisome, since reverse causality has the potential to bias model fits and produce spurious results. Even if there was no causal effect of density, we might observe statistically significant density effects in our models. For example, suppose that due to other unmeasured covariates or random chance, some days have a higher than average rate of fish passage into a fishway. On those days, we would observe a higher count, and empirically the rate of passage would positively correlate with count, despite having no causal effect of density on passage.

We took two major steps to account for these modelling challenges. First, although hourly count data were available, we used daily count data to estimate density. Since a consistently high proportion of radio-tagged fish in our dataset passed within 24 h (greater than 97%), and relatively few fish passed overnight (less than 10% entering the fishway and less than 5% exiting to the reservoir), the daily count provided a roughly accurate assessment of daily density while smoothing out any within-day variation in passage rate. On the other hand, hourly counts are far more dependent on random chance and are systematically biased due to daily variations in passage rate. Moreover, hourly counts increase the

risk of other biases due to the time lag between a tagged fish triggering a telemetry event and being counted by an observer.

Second, we used a parametric bootstrap to protect against reverse causality. Our bootstrap analysis was justified using the following chain of logic. First, our system violated modelling assumptions and our models may therefore produce biased inference. Second, if density truly has no effect then counts have no effect. In this case, we need not include counts as a covariate, and our models should provide valid unbiased fits. Therefore, simulations from models with no count covariates produce valid null distributions. By fitting models *with* count covariates to these null model simulations, we obtained a null distribution for the count coefficient. Even if the above problems induced reverse causality in our dataset, our null distribution remains valid, since the null model has no reverse causality. Due to reverse causality, the power of our tests remains unknown, but we may nevertheless correctly calculate *p*-values. For a more detailed introduction to bootstraps in general see Efron & Tibshirani [42]. Generally, we use *null model* to mean a model without a count covariate.

## (d) Statistical analysis

To satisfy model assumptions, we split our initial four PH models (2 species × 2 processes) into 11 models. We did not split our two logistic regression models. By splitting a model, we mean we split the dataset for that model into disjoint subsets and fit separate but structurally identical models to each set of subsets. We split all PH (finding; fishway) models by dam (TD; JD). We further split our Chinook fishway models by run (spring; summer) and our sockeye fishway TD model by specific fishway (east; north). These submodels are referenced in table 1 and are explained in further detail in the electronic supplementary material. One PH model (sockeye fishway TD north) was discarded due to low sample size, leaving us with 12 final models. In each of those 12 models, we conducted one set of null model simulations (bootstraps) to test for the presence of

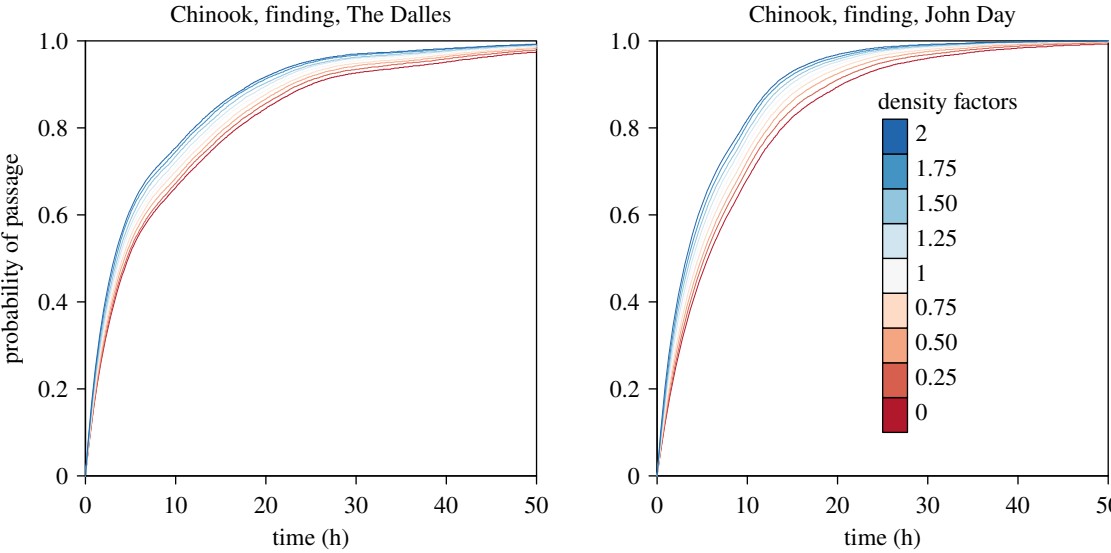

**Figure 2.** Predictions from our two Chinook 'finding' models under various fish density scenarios. Densities were chosen to be a factor (see legend) multiplied against measured counts, to simulate a realistic scenario of higher or lower densities. Factors ranged from 0, to simulate near-extirpation up to twice current levels. Curves at each density factor were generated by simulating from the fitted model 192 times and calculating the median probability of passage at each time point. (Online version in colour.)

a density effect as well as a set of secondary bootstraps to calculate confidence intervals for the count coefficients.

After conducting model diagnostics, we added a threshold density effect (count $\leq 150$) to our Chinook finding model. This successfully accounted for a series of large positive residuals, possibly modelling a strong crowding effect (i.e. negative density effect). However, this effect was highly confounded with fishway identity, with 88 out of 89 fish under the threshold passing at TD's east fishway, and because of this no strong conclusions can be made about this effect. We used this effect as part of the null model, prior to testing for a linear density effect.

To ensure that our fitted density effects were biologically important as well as statistically significant, we produced simulations for a variety of density scenarios ranging from zero density (near-extirpation) to twice current density, which can be found in figure 2. Each density scenario is obtained by multiplying observed counts by a fixed factor, and then producing model simulations. We did this for both the PH models and the logistic regression models.

For additional technical details regarding our statistical analysis, please refer to the electronic supplementary material.

## 3. Results

Of our 12 tests, three detected significant ($p < 0.001$) density effects. Two were Chinook PH models for finding the fishway, and the third was a Chinook logistic regression model for committing to a fishway. Two other effects were significant at the $p < 0.05$ or $0.1$ levels (summer-run Chinook navigating the TD fishway; and sockeye navigating the TD east fishway). However, since these two effects were statistically weak and did not display any consistent pattern with other models of the same species and process, we concluded that these associations were spurious. All significant density effects were positive, meaning that an increase in density facilitated faster predicted completion of the modelled process, and therefore faster overall passage.

Our simulated null distributions for the count coefficients are shown in figure 3, compared to the actual count coefficient

fitted to our true (non-simulated) dataset. These results along with our confidence intervals, are shown in table 1. Coefficients are reported after standardizing all covariates so that magnitudes can be compared. Full model summaries are available in the electronic supplementary material.

These results provide strong evidence for the existence of positive density effects, particularly in Chinook salmon. No evidence was found for negative density effects. Evidence for sockeye salmon did not support a density effect.

Our density scenario simulations for the 'finding' models can be found in figure 2. These simulations predict a 20–30% increase in finding times for Chinook under the extirpation scenario compared to present conditions, and 50–115% more fish with high finding times of greater than 24 h. Conversely, if salmon densities were to double compared to 2013/2014 levels we predicted a 15–19% decrease in finding times and 31–46% fewer fish with high (>24 h) finding times. Our density scenario simulations for the 'committing' model predicted a linear increase in commit rates with densities, with doubling densities bringing an increase of almost 5% commit rate from the current level of 43% to 47.7%. However, lower density predictions from this model were unreliable, due to the threshold density effect in our null model. Because this effect was only evident in one model, and was confounded with fishway identity we were not able to fully investigate this possible density effect, which has a profound effect on our predictions at low densities.

Consistent with previous studies in the same system [29,43], we also observed strong temperature ($0.22 < |\beta| < 1.29$; negative in 2/4 Chinook fishway models, otherwise positive) and very strong diel ($1.22 < \beta < 2.96$; higher passage during the day) effects in our PH models. We observed a strong temperature effect ($\beta = -1.23$) in our Chinook committing model. Our significant density coefficients were of roughly the same order of magnitude as our temperature coefficients, indicating that density may have a substantial impact, comparable to other established drivers. Since these other variables were not our focus, we did not conduct any further analysis of these environmental effects. Model summaries are available in the electronic supplementary material.

Proc. R. Soc. B 287: 20202137

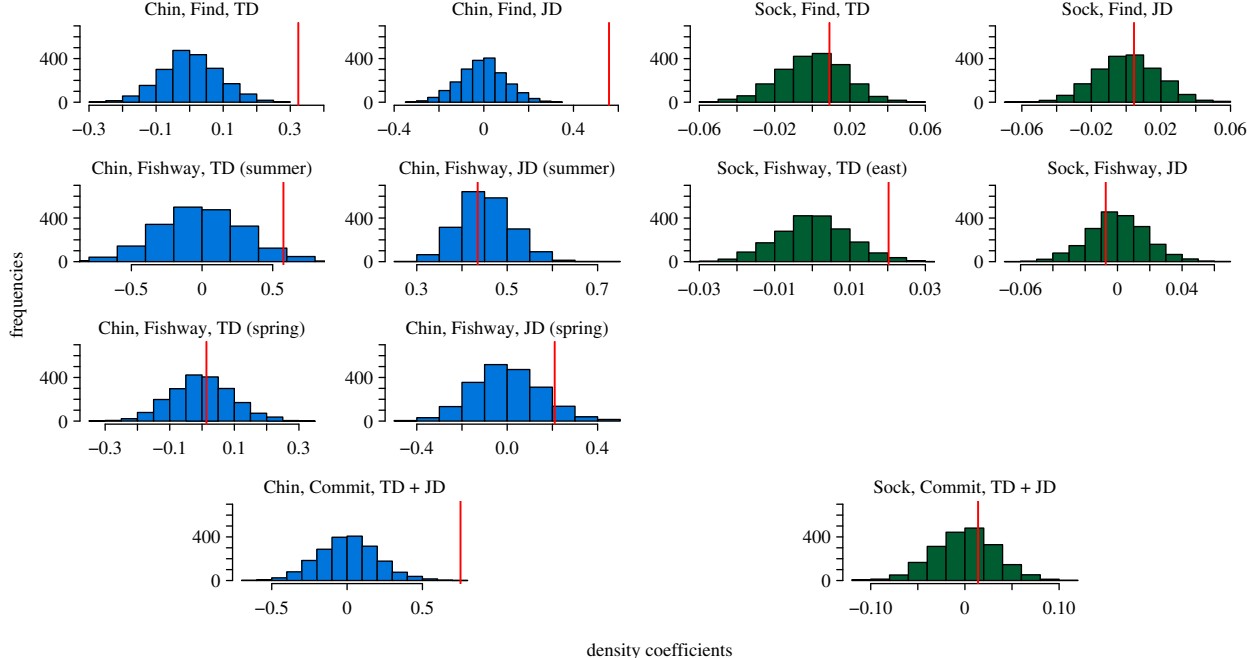

**Figure 3.** The results of our null distribution simulations. Histograms represent fitted density coefficients from over 2000 simulations of the best fitting null model (with no density effect). Vertical lines represent the density coefficient fitted to our actual data. Chin refers to models of Chinook salmon; Sock refers to models of sockeye salmon. Find, Fishway and Commit refer to our three different process models (figure 1). TD and JD refer to The Dalles and John Day Dams. summer and spring refer to models in which summer-run and spring-run Chinook were separated. East refers to the east fishway at The Dalles Dam. Separation of models beyond species and process were the result of model diagnostic procedures. (Online version in colour.)

## 4. Discussion

Due to human modifications of the global landscape, animals face increased migratory challenges [44]. Salmon returning to spawning grounds in the Columbia River basin are confronted with multiple main stem dams. While many dams have fishways, passage remains a daunting challenge as fish need to locate a fishway entrance and then ascend a ladder to continue their migration. Here, we reveal evidence that Chinook, but not sockeye, salmon appear to benefit from social interactions during this challenge. We demonstrated, using a bootstrap method to account for several sources of reverse causality, a positive effect of density on two key quantities for Chinook salmon: rates of locating fishway entrances, and probability of committing to passage of a fishway. By contrast, we found no density effect for passage rates through a fishway, or for any aspect of dam passage for sockeye salmon.

Our results can alternatively be explained by an effect of migratory motivation or run-timing, as other studies have noted [30]. Fish on peak density days may be more motivated, or may have timed their migration better with respect to unmeasured environmental conditions than earlier- or later-migrating fish. Controlling for these effects without directly manipulating density is difficult. Given the consistency of variation between species, we feel that the most plausible explanation is the presence of a true density effect in Chinook salmon, rather than an effect of motivation or timing, which we might expect to see more strongly in both species. Generally, we see our lack of statistically 'significant' results for sockeye salmon as adding support to our results for Chinook salmon. We do not expect density effects to exist in all processes or for all species, but do expect these effects to apply consistently.

It is nevertheless counterintuitive that the species with a stronger tendency to school might benefit less from collective

navigation. Here, we provide two plausible, yet speculative, explanations. First, having a strong tendency to migrate in groups [45], sockeye salmon might form sizeable groups regardless of population density—i.e. their local density may not be highly dependent on global density. By contrast, the less-social Chinook salmon's local density might be more governed by global density. Thus for Chinook the number of social interactions may scale with daily fish density whereas for sockeye social interactions may be relatively constant. Second, casual observations by the authors suggest another potential mechanism. In tailraces, Chinook salmon tend to hold near fishway entrances before entering. This holding behaviour seems to result in loose shoal-like aggregations near fishway entrances, potentially leaving a sort of social signpost that may draw subsequent Chinook towards an entrance. On the other hand, sockeye salmon's increased tendency to school might inhibit their ability to take advantage of such a social effect (as in Lemasson *et al.* [46]). By travelling in large, cohesive, polarized schools, sockeye salmon may have trouble transitioning from a large tailrace to a relatively small fishway. The social 'momentum' of the school may prevent individuals who do spot a fishway entrance from stopping to explore it further. This possible mechanism also explains why we observed no strong effects within the fishways, although it does not explain the effect we observed for commit probabilities.

The effect of density on fishway passage rates was first studied over 60 years ago, but has largely focused on the effects of overcrowding (i.e. deleterious effects of density). The possibility of a positive effect of density on fishway passage rates was discussed by Lander [47] in the context of a model for overcrowding, and saw experimental work in a study of alewives (*Alosa pseudoharengus*), also in the context of over-crowding [48]. Unfortunately, although Dominy [48] reported a positive density effect, they analysed their data in a way

that was inconsistent with our analysis.[1] More recent work in the context of culvert passage has found negative [21], and no [49] density effects for brook trout (*Salvelinus fontinalis*) and coho salmon (*O. kisutch*), respectively. These study systems were most similar to our 'fishway' system, involving little difficulty in finding the entrance to the culvert (e.g. in both studies fish were confined at the downstream extremity, as opposed to our open system). Thus our null results for 'fishway' are consistent with Johnson *et al.* [49], while our positive result for 'finding' involves a fundamentally different, relatively unstudied system. A study by Caudill *et al.* [30] on Chinook salmon and steelhead (sea-run *O. mykiss*) reported several positive associations with density, but density effects were not the focus of their analysis, and their model encompassed the entire dam passage process from tailrace entry to fishway exit. A study of juvenile palmetto bass (*Morone saxatilis × chrysops*), Lemasson *et al.* [46] found that fish in schools took much longer than lone individuals to pass an artificial barrier when moving downstream and thus showed a negative effect of density on passage rate.

Given the diversity of findings in this literature, and in our own results, there is scope for additional work in this area. Our methods can be applied to other systems and species, where 'finding' and 'committing' behaviours have rarely been isolated for analysis. For example, controlled navigation experiments investigating these behaviours, where density can be manipulated systematically, are a promising avenue. Emerging technologies including sonic tags, acoustic cameras, and computer vision will make more detailed analysis of individual and collective movement around dams possible and also yield local (i.e. actual group size, rather than estimated fish density in tailraces) measures of conspecific density [50]. Such studies could also shed light on the specific mechanisms driving any collective navigation. Revealing individual and collective search algorithms may contribute general principles to the fields of animal movement and bioinspired engineering, but may also contribute insights for dam management of fish passage. For example, mechanisms such as the 'social signpost' hypothesis for Chinook salmon might motivate management interventions designed to promote more efficient salmon migration past fishways. One simple hypothetical intervention might simulate a holding pattern of fish about fishway entrances using model fish. These decoys could provide a social signpost, attempting to activate our hypothesized social behaviours even at low densities.

When conducting our analysis we added an additional threshold density effect to our Chinook 'commit' model, possibly modelling a strong crowding effect. However, we did not conduct an in-depth analysis of this variable, since it was confounded with use of the TD east fishway. One possible explanation for this effect is as a density effect associated with overcrowding in the fishway—however, it is unclear why overcrowding would be represented by a threshold effect rather than a more gradual decline in commit probabilities. Furthermore, even if this density effect were associated with overcrowding in one particular fishway, we would expect such an effect to vary among fishways, making it difficult to generalize this effect to make predictions elsewhere. We recommend this as an avenue for further research.

Our analysis faced three substantial challenges: counts were confounded with hazard, counts were a time-delayed estimate of density, and density was confounded with hazard across time. We addressed these challenges using our bootstrap approach to circumvent concerns about reverse causality, and by using smoother daily counts rather than hourly counts which are more subject to concerns about time-delay and inconsistent relationships with density. These were the more unique roadblocks we encountered, but like any statistical analysis, there are some other caveats to consider. For example, other covariates such as flow velocity may have played an important role but were not included, and some variables such as migratory motivation are all but impossible to measure quantitatively in any case. Like any model, ours was an imperfect representation of reality. We feel it was close enough to reality to provide useful insight. Another caveat of our analysis is that we dropped fish with missing measurements from our dataset, rather than conducting a so-called 'censoring analysis' of missing data. Since we used only 'entry' and 'exit' data points, a proper censoring analysis was impossible. Furthermore, since data points were likely missing independently at random due to missed radiotelemetry signals, we incurred little bias in this manner.

To our knowledge, our results provide the first demonstration of a positive density effect for fish passing riverine obstacles. This is in contrast to the prevailing wisdom that overcrowding is the dominant effect. More broadly, it is one of only a handful of examples of collective navigation in freely-migrating populations. Given the ubiquity of social movement during migration, we expect many more examples of collective navigation to be uncovered, especially as new technologies improve our ability to quantify collective movement *in situ* [50]. Such studies underscore the need for further investigation of density effects, since these processes could have important ecological implications. For example, if populations decline, densities at dams will decline: will this alleviate overcrowding or reduce collective benefits? In the first case, population decline is buffered, while in the other it is magnified, potentially generating critical transitions that may lead to sudden population collapses [15,51]. As anthropogenic disturbances simultaneously increase navigational difficulty in a variety of contexts and decrease population densities, understanding the role of density-dependent processes, such as collective navigation, may therefore yield critical insights for sound management and conservation [52].

Data accessibility. All data used in this article are from publicly available repositories. All data for this article have been made publicly available on figshare at https://doi.org/10.6084/m9.figshare.13010072.v1, along with links to source datasets and R code which reproduces our data preprocessing, analysis, model diagnostics, and results.

Authors' contributions. A.B. and P.W. conceived the original idea, M.K. collected and provided the data, C.O. designed and conducted the analyses, and all authors helped write the manuscript.

Competing interests. We declare we have no competing interests.

Funding. Telemetry data were collected with funding from the US Army Corps of Engineers, Portland District and with assistance from the NW Fisheries Science Center (NOAA-Fisheries) and the University of Idaho Fish Ecology Research Lab. This material is based upon work supported by the NSF-GRFP under grant no. DGE-1762114.

Acknowledgements. We thank David Smith and Christa Woodley for assistance obtaining data and to Christy Contreras for preliminary analysis as part of an NSF REU project at the Santa Fe Institute. Finally, we thank Chris Caudill for invaluable support.

## Endnote

[1]Dominy [48] used a numerical response (number passing per unit time) rather than the rate response (proportion passing per unit

time) used here. Although they reported a sigmoidal (positive at low densities, then negative at high densities) density effect in alewives, a visual analysis of their data appears more consistent (in our framework) with either no density effect or a negative effect, although there is a large group of outliers that could indicate a positive effect.

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
