## [Reviewer comments · Proceedings of the Royal Society B: Biological Sciences]

Review History

RSPB-2020-0624.R0 (Original submission)

Review form: Reviewer 1

Recommendation

Accept with minor revision (please list in comments)

Scientific importance: Is the manuscript an original and important contribution to its field?

Excellent

General interest: Is the paper of sufficient general interest?

Good

Quality of the paper: Is the overall quality of the paper suitable?

Excellent

Is the length of the paper justified?

Yes

Should the paper be seen by a specialist statistical reviewer?

No

Do you have any concerns about statistical analyses in this paper? If so, please specify them explicitly in your report.

No

It is a condition of publication that authors make their supporting data, code and materials available - either as supplementary material or hosted in an external repository. Please rate, if applicable, the supporting data on the following criteria.

Is it accessible?

Yes

Is it clear?

Yes

Is it adequate?

Yes

Do you have any ethical concerns with this paper?

No

Comments to the Author

In the manuscript titled "Collective navigation facilitates passage through human-made barriers by homeward migrating Pacific salmon" the authors describe an experimental study regarding anthropogenic effects on animal behavior in the wild, more specifically collective migration of salmon through dams. This challenging problem was tackled by neatly combining benefits of two different datasets (on coarse and fine scales) and rigorous statistical analysis. The results show that navigating collectively indeed might help salmon overcome these barriers in some cases. This could be of great importance for future management and monitoring on this type of wild life - potentially an important contribution to conservation biology - as well as give quantitative insight on how to design fish ladders that have the least possible negative impact on salmon populations. Therefore I think this paper is of broad interest also for researchers from multiple, different fields. That said - and while the manuscript is generally well written - I think the paper in its current form is mostly accessible for readers specialized to this field based on its terminology (especially in section Statistical Analysis), and this could be improved by carefully keeping in mind a broader perspective. So parts of the manuscript should be re-written to apply better for a wide readership.

The results and conclusions are carefully thought through and interpreted, with alternatives being discussed. I think some parts could be improved by making some clarifications and correcting small issues. For instance, the figures offer no clear way to judge the main results (or conclusions), and Figure 3 isn't referred to in the text.

More specific comments:

It is great value of the manuscript that the authors provide a full disclosure with the results, but currently what is lacking is a more thorough discussion as to what might lead to the difference seen between the two species, and also the difference between this work with previous studies showing in some cases opposite results.

Adding a section in the discussion about pitfalls of these types of analyses, where you dealt with many potential parameters at play, and there are limitations to our ability to measure them all. What might be a good future direction? More work in controlled lab conditions? What is the state of the art in measurement devices that we could expect to see utilised soon in the field?

To account for readers from different fields, it would be convenient if terminology of "crowding" vs. "density" would be made clear.

Please write out specialized abbreviations at their first use like DART (data access in real time),

As mentioned before, I think using the combination of two different types of measurements is a great approach in this manuscript, but the differences between what DART and telemetry provide (advantages and drawbacks) should be explained in a bit more detail (especially for non-specialist readers), and therefore what you gain by using both.

Could there be learning effects between dams? Can you comment on this?

As a consequence of all above, I suggest acceptance after minor revision. As the changes suggested are minor, the Editor may decide to accept them without my further involvement. But I am also happy to check the revised version of this manuscript.

Review form: Reviewer 2

Recommendation

Major revision is needed (please make suggestions in comments)

Scientific importance: Is the manuscript an original and important contribution to its field?

Good

General interest: Is the paper of sufficient general interest?

Good

Quality of the paper: Is the overall quality of the paper suitable?

Acceptable

Is the length of the paper justified?

No

Should the paper be seen by a specialist statistical reviewer?

No

Do you have any concerns about statistical analyses in this paper? If so, please specify them explicitly in your report.

No

It is a condition of publication that authors make their supporting data, code and materials available - either as supplementary material or hosted in an external repository. Please rate, if applicable, the supporting data on the following criteria.

Is it accessible?

Yes

Is it clear?

Yes

Is it adequate?

Yes

Do you have any ethical concerns with this paper?

No

Comments to the Author

It was a pleasure reviewing this manuscript on a topic that have not been widely studied so far. I am suggesting a major revision and the inclusion of data on other species if possible. Please find detailed comments in the attached document.

Decision letter (RSPB-2020-0624.R0)

01-May-2020

Dear Mr Okasaki:

I am writing to inform you that your manuscript RSPB-2020-0624 entitled "Collective navigation facilitates passage through human-made barriers by homeward migrating Pacific salmon" has, in its current form, been rejected for publication in Proceedings B.

This action has been taken on the advice of referees, who have recommended that substantial revisions are necessary. With this in mind we would be happy to consider a resubmission, provided the comments of the referees are fully addressed. However please note that this is not a provisional acceptance.

Sincerely,
Professor Gary Carvalho
<mailto:proceedingsb@royalsociety.org>

Associate Editor
Board Member: 1
Comments to Author:

I have enjoyed this paper and likewise the two referees. Nonetheless, there are many opportunities to improve the paper - particularly the methods could be more accessible to non-specialists and the rationale for the parametric bootstrap and its effectiveness needs to be more clearly explained and justified. Referee 2 has numerous comments re the methods and results that

you should consider, and I agree that the Figure 1 could be more descriptive of the data and that the presentation of the results is too brief and needs to be fleshed out. The results by bullet points are an odd choice and I would prefer full text explaining those results, and perhaps a table to organize the statistics. The discussion could be expanded as well, particularly missing is a discussion of caveats, limitations, and assumptions of the analyses (data and models). I would also offer that the discussion is too enthusiastic of what is obviously the favoured hypothesis and there is no consideration that half the results (one of the two species) do not support the hypothesis - why is that? why do the species differ? is that a true biological difference (if so how?) or a data limitation? Finally, note that I disagree with referee 2 that the addition of more species in the analysis is essential. It would certainly help, and if such data exist from the same dams why not model them too? Alternatively an explanation for why that was not done would be helpful.

Reviewer(s)' Comments to Author:

Referee: 1

Comments to the Author(s)

In the manuscript titled "Collective navigation facilitates passage through human-made barriers by homeward migrating Pacific salmon" the authors describe an experimental study regarding anthropogenic effects on animal behavior in the wild, more specifically collective migration of salmon through dams. This challenging problem was tackled by neatly combining benefits of two different datasets (on coarse and fine scales) and rigorous statistical analysis. The results show that navigating collectively indeed might help salmon overcome these barriers in some cases. This could be of great importance for future management and monitoring on this type of wild life - potentially an important contribution to conservation biology - as well as give quantitative insight on how to design fish ladders that have the least possible negative impact on salmon populations. Therefore I think this paper is of broad interest also for researchers from multiple, different fields. That said - and while the manuscript is generally well written - I think the paper in its current form is mostly accessible for readers specialized to this field based on its terminology (especially in section Statistical Analysis), and this could be improved by carefully keeping in mind a broader perspective. So parts of the manuscript should be re-written to apply better for a wide readership.

The results and conclusions are carefully thought through and interpreted, with alternatives being discussed. I think some parts could be improved by making some clarifications and correcting small issues. For instance, the figures offer no clear way to judge the main results (or conclusions), and Figure 3 isn't referred to in the text.

More specific comments:

It is great value of the manuscript that the authors provide a full disclosure with the results, but currently what is lacking is a more thorough discussion as to what might lead to the difference seen between the two species, and also the difference between this work with previous studies showing in some cases opposite results.

Adding a section in the discussion about pitfalls of these types of analyses, where you dealt with many potential parameters at play, and there are limitations to our ability to measure them all. What might be a good future direction? More work in controlled lab conditions? What is the state of the art in measurement devices that we could expect to see utilised soon in the field?

To account for readers from different fields, it would be convenient if terminology of "crowding" vs. "density" would be made clear.

Please write out specialized abbreviations at their first use like DART (data access in real time),

As mentioned before, I think using the combination of two different types of measurements is a great approach in this manuscript, but the differences between what DART and telemetry provide (advantages and drawbacks) should be explained in a bit more detail (especially for non-specialist readers), and therefore what you gain by using both.

Could there be learning effects between dams? Can you comment on this?

As a consequence of all above, I suggest acceptance after minor revision. As the changes suggested are minor, the Editor may decide to accept them without my further involvement. But I am also happy to check the revised version of this manuscript.

Referee: 2

Comments to the Author(s)

It was a pleasure reviewing this manuscript on a topic that have not been widely studied so far. I am suggesting a major revision and the inclusion of data on other species if possible. Please find detailed comments in the attached document.

Author's Response to Decision Letter for (RSPB-2020-0624.R0)

See Appendix A.

RSPB-2020-2137.R0

Review form: Reviewer 2

Recommendation

Accept with minor revision (please list in comments)

Scientific importance: Is the manuscript an original and important contribution to its field?

Excellent

General interest: Is the paper of sufficient general interest?

Excellent

Quality of the paper: Is the overall quality of the paper suitable?

Excellent

Is the length of the paper justified?

Yes

Should the paper be seen by a specialist statistical reviewer?

No

Do you have any concerns about statistical analyses in this paper? If so, please specify them explicitly in your report.

No

It is a condition of publication that authors make their supporting data, code and materials available - either as supplementary material or hosted in an external repository. Please rate, if applicable, the supporting data on the following criteria.

Is it accessible?

N/A

Is it clear?

N/A

Is it adequate?

N/A

Do you have any ethical concerns with this paper?

No

Comments to the Author

It was a pleasure to review this manuscript again. Thank you for providing thoughtful responses to my initial comments, and for revising the manuscript accordingly. I find it much improved and easier to read. I also very much like your updated Figure 1, as well as the new Figure 3. I have a few minor comments and suggestions that are listed in the attached document.

Decision letter (RSPB-2020-2137.R0)

23-Sep-2020

Dear Mr Okasaki

I am pleased to inform you that your manuscript RSPB-2020-2137 entitled "Collective navigation facilitates passage through human-made barriers by homeward migrating Pacific salmon" has been accepted for publication in Proceedings B.

The referee(s) have recommended publication, but also suggest some minor revisions to your manuscript. Therefore, I invite you to respond to the referee(s)' comments and revise your manuscript. Because the schedule for publication is very tight, it is a condition of publication that you submit the revised version of your manuscript within 7 days. If you do not think you will be able to meet this date please let us know.

[http://datadryad.org/submit?journalID=RSPB&manu=\(Document not available\)](http://datadryad.org/submit?journalID=RSPB&manu=(Document+not+available)) which will take you to your unique entry in the Dryad repository. If you have already submitted your data to dryad you can make any necessary revisions to your dataset by following the above link. Please see <https://royalsociety.org/journals/ethics-policies/data-sharing-mining/> for more details.

Sincerely,
Professor Gary Carvalho
mailto: proceedingsb@royalsociety.org

Associate Editor

Comments to Author:

The revised manuscript is much improved. There remain a few minor suggestions from the referee. I also recommend the final sentence of the abstract needs to be changed from "...in at least some species,..." to "...in at least one species,..." as the positive results are for Chinook salmon only.

Reviewer(s)' Comments to Author:

Referee: 2

Comments to the Author(s).

It was a pleasure to review this manuscript again. Thank you for providing thoughtful responses to my initial comments, and for revising the manuscript accordingly. I find it much improved and easier to read. I also very much like your updated Figure 1, as well as the new Figure 3. I have a few minor comments and suggestions that are listed in the attached document.

Author's Response to Decision Letter for (RSPB-2020-2137.R0)

See Appendix B.

Decision letter (RSPB-2020-2137.R1)

30-Sep-2020

Dear Ms Okasaki

I am pleased to inform you that your manuscript entitled "Collective navigation can facilitate passage through human-made barriers by homeward migrating Pacific salmon" has been accepted for publication in Proceedings B.

Open Access

Paper charges

Sincerely,

Appendix A

Response to Reviewers

Colin Okasaki, Matthew Keefer, Peter Westley, Andrew Berdahl

August 29, 2020

We would like to thank the editor for arranging such constructive and thoughtful reviews of our manuscript and although they have identified some concerns, we were pleased with their generally favorable view of our work. Here we respond to each comment in turn. Throughout, comments from the editor and reviewer are highlighted in blue. Our response is written in black.

1 Response to the Editor

I have enjoyed this paper and likewise the two referees. Nonetheless, there are many opportunities to improve the paper - particularly the methods could be more accessible to non-specialists

See our new section Modeling Approach which now provides a more conceptual discussion of our methods, why they were necessary, and why they work. In particular, see lines 146–174 where we introduce the basics of time-to-event modeling, and lines 178–188 where we have attempted to make our exposition more accessible by explaining it entirely conceptually, without relying on equations. Although we hoped to address other comments and produce a more balanced manuscript, our edits towards accessibility necessarily lengthened our methods section further. To address this and to further make our main text more accessible, we moved most of the necessarily technical detailed discussion of our statistical analysis to supplementary materials. For this reason, we left ESM lines 8–100 largely unchanged. We left only the bare minimum in the section Statistical Analysis to contextualize our results.

and the rationale for the parametric bootstrap and its effectiveness needs to be more clearly explained and justified.

See our edits to lines 189–197 which discuss the problems posed by reverse causality, which necessitated the bootstrap approach. See also our edits to lines 207–218 which discuss how the bootstrap solves these problems. In short, reverse causality breaks model assumptions and may result in unforeseen model biases. The bootstrap approach produces unbiased null distributions so that we may calculate p -values; however, due to the reverse causality we may not calculate the power of our tests.

Referee 2 has numerous comments re the methods and results that you should consider, and I agree that the Figure 1 could be more descriptive of the data and that the presentation of the results is too brief and needs to be fleshed out.

We have modified both Figure 1 and its caption. In particular, we have included in Figure 1 the telemetry array as requested by Reviewer 2, and in the caption we have included additional measurements relevant to our study, particularly the distances from the tailrace antennas to the dams.

We have expanded the text of our Results section somewhat, and have included a table for our numerical results and a new figure to contextualize our effect sizes. Our Results section is still necessarily brief compared to our Methods – we used a fairly complex modeling approach to arrive at what is a fairly straightforward numerical result. However, we have also expanded the text of our Discussion section dramatically, adding paragraphs for a roadmap, discussion of alternative explanations for our results, discussion of possible mechanisms underlying our results, and future work. We hope that this sufficiently addresses the concerns regarding our paper's balance.

The results by bullet points are an odd choice and I would prefer full text explaining those results, and perhaps a table to organize the statistics.

We have reformatted our results into Table 1 and agree that this is a clearer presentation. This also allowed us to incorporate more information, such as sample size.

The discussion could be expanded as well, particularly missing is a discussion of caveats, limitations, and assumptions of the analyses (data and models).

We have expanded the discussion dramatically, but see in particular lines 362–376 where we have added a paragraph discussing some of the most important caveats associated with our analyses.

I would also offer that the discussion is too enthusiastic of what is obviously the favoured hypothesis and there is no consideration that half the results (one of the two species) do not support the hypothesis – why is that? why do the species differ? is that a true biological difference (if so how?) or a data limitation?

We have added discussion of the differences between species and between systems in our results and provide two plausible behavioural mechanisms to explain these differences (lines 282–314).

We do believe this difference to be a true result (as opposed to a data limitation). With our updated methods, we have fewer significant results but we believe that we demonstrate these results more convincingly. We have included one hypothesis as to why the species might differ in their responses, and also have noted in lines 321–328 how the pattern of positive and null results is consistent with previous research in the similar area of culvert passage.

Finally, note that I disagree with referee 2 that the addition of more species in the analysis is essential. It would certainly help, and if such data exist from the same dams why not model them too? Alternatively an explanation for why that was not done would be helpful.

We agree with referee 2 that analyzing this sort of data with more species and more systems is a valuable next step. We took this feedback very seriously, but nevertheless concluded that including additional data was beyond the scope of our study. As we see it there are four types of additional data that might have been included: data from additional systems, species, dams, or years. We did not include such data in this analysis for the following reasons.

Firstly the telemetry dataset we worked with is quite unique, and few comparable data sets in other systems exist to synthesize with our own. What data sets may exist are likely different in many ways that would obfuscate statistical analysis. We have pieced together data from several different sources to conduct our analysis, but each covariate was pulled consistently from a single source, which lends confidence that each covariate has a consistent and stable interpretation. This is lost once we begin to include data from other existing projects. Follow up work in new systems could certainly involve collection of new telemetry data, but that was certainly outside the scope of this project.

Staying within the telemetry data set that we analyzed, we could have included data from other dams and other species, however these posed further problems. In some cases, other species pose quite different questions: for example, steelhead (anadromous *O. mykiss*) are partially iteroparous, and this might contribute to a confounding learning or leadership effect. Other species, such as American shad, were telemetry tagged only on an ad hoc basis, and small sample sizes would plague our analysis, which relies on large sample sizes to fit complex models. Furthermore, we have the most experience with salmon biology and behaviour and would have a much more difficult time interpreting results from shad or other such species.

Similar problems prevented extension of our model to additional dams. We chose to exclude the first mainstem dam, Bonneville, because among other issues (e.g. sea lion predation) this is where salmon were captured for tagging; salmon were released only a short distance downstream, resulting in correlated arrival times among tagged fish. The fourth mainstem dam, McNary, might have been suitable for analysis, but later dams would involve analyzing datasets with smaller and smaller sample sizes. As we saw in our analysis, our models become difficult to fit somewhere between $n \approx 300$ and $n \approx 60$. Adding additional dams would involve fitting an increasingly piecemeal collection of models split to subsets of subsets of our data, potentially clouding interpretation of any results.

Finally, we had in our telemetry dataset other years we could have analyzed. We analyzed data from 2013 and 2014. The only other year in which both Chinook and sockeye were sampled was 1997. We excluded this year because not all of our environmental covariates were available, and because we are leery of inappropriately re-using old data from the same

collaboration that has previously hinted at density effects (i.e. Caudill et al 2007). Including data from only some species in some years risks confounding our results with differing annual conditions.

Of all these limitations, it would have been easiest to include data from non-matching years, or to include data from McNary dam. However, as Referee 2 seems to note, the primary value in adding data to this analysis would come from analyzing additional species or systems. Our analysis of Chinook and sockeye salmon at The Dalles and John Day dams in 2013 and 2014 is sufficiently robust to produce what we believe to be convincing statistical conclusions. Adding further data of such strictly analogous types would likely reduce our p -values, but we believe it would accomplish little else, while coming at a substantial cost in additional time, energy, and computational resources.

2 Response to Referee 1

In the manuscript titled Collective navigation facilitates passage through human-made barriers by homeward migrating Pacific salmon the authors describe an experimental study regarding anthropogenic effects on animal behavior in the wild, more specifically collective migration of salmon through dams. This challenging problem was tackled by neatly combining benefits of two different datasets (on coarse and fine scales) and rigorous statistical analysis. The results show that navigating collectively indeed might help salmon overcome these barriers in some cases. This could be of great importance for future management and monitoring on this type of wild life - potentially an important contribution to conservation biology - as well as give quantitative insight on how to design fish ladders that have the least possible negative impact on salmon populations. Therefore I think this paper is of broad interest also for researchers from multiple, different fields. That said - and while the manuscript is generally well written - I think the paper in its current form is mostly accessible for readers specialized to this field based on its terminology (especially in section Statistical Analysis), and this could be improved by carefully keeping in mind a broader perspective. So parts of the manuscript should be re-written to apply better for a wide readership.

We feel that this feedback was particularly valuable. Although mindful of other feedback we received regarding balance within the manuscript (i.e. an overly long methods section) we took this feedback and expanded our Statistical Analysis section into two sections. The first section, Modeling Approach, we tried to make more accessible, using “time-to-event” terminology rather than “survival” and adding an extra paragraph explaining the basics of this modeling framework. We feel it is important to include the mathematical underpinnings of time-to-event modeling but have tried to rephrase to conceptual language when possible in lieu of equations. We have also relegated much of the more technical statistical methods to the new section Statistical Analysis. In this way an interested reader without much mathematical background might skip, say, from line 161 to line 241, while an interested reader with a foundation in mathematics but not in statistics or time-to-event modeling might read the whole paper but not the ESM. We feel that with our new expanded results and discussion sections these readers should be able to obtain the major takeaways of our article and we hope this represents a satisfactory balance between accessibility and mathematical rigor.

The results and conclusions are carefully thought through and interpreted, with alternatives being discussed. I think some parts could be improved by making some clarifications and correcting small issues. For instance, the figures offer no clear way to judge the main results (or conclusions), and Figure 3 isn't referred to in the text.

We have added two new figures to clarify our results and analysis. First, we added Figure 1 in our new ESM to better communicate our model splitting and how we arrived at the rather complex set of models we ultimately fit. Second, we added Figure 3 to communicate the effect size of our fitted density effects. We feel that our main results are: (1) we found evidence for density effects, despite the statistical gymnastics required by potential reverse causation; we hope this is communicated in Figure 2 and (2) these density effects are not just

an artifact of large sample sizes, but in fact represent biologically meaningful and important difference in passage rates; we hope this is communicated by our new Figure 3.

More specific comments: It is great value of the manuscript that the authors provide a full disclosure with the results, but currently what is lacking is a more thorough discussion as to what might lead to the difference seen between the two species

We were initially hesitant to discuss this difference from a mechanistic standpoint because our results are so purely statistical. Strictly speaking we can interpret our models only as showing i.e. “Chinook find the fishways faster on higher density days.” We did not conduct any analysis capable of really distinguishing different mechanisms from one another. However, we have now included some speculation as to what might cause these differences in lines 282–314. Many mechanisms are possible, and we feel that this certainly deserves further research.

and also the difference between this work with previous studies showing in some cases opposite results.

We have added additional text in lines 321–328 to explain the differences between our work and previous studies. We believe when examining our results by mechanism, that the fishway process (where we observed largely null results) aligns most closely with previous work that has found null and negative results. On the other hand the finding and committing processes, where we saw evidence for strong positive density effects, are more poorly studied (at least where density is concerned; these processes have been well-studied with respect to other variables). See also lines 80–81 where we note that none of the prior studies which found negative effects included sockeye or Chinook salmon.

Adding a section in the discussion about pitfalls of these types of analyses, where you dealt with many potential parameters at play, and there are limitations to our ability to measure them all.

We have added lines 362–376 to the discussion where we acknowledge several of the major concerns we, and others, may have about our analysis.

What might be a good future direction? More work in controlled lab conditions? What is the state of the art in measurement devices that we could expect to see utilised soon in the field?

Controlled experiments and emerging technologies allowing for *in-situ* movement recording both have tremendous potential to push our understanding forward here. We have added a half paragraph in the discussion (lines 334–345) to serve as a road map for this future research and outline its potential importance.

To account for readers from different fields, it would be convenient if terminology of crowding vs. density would be made clear.

By “crowding” we meant to refer to negative or deleterious effects of fish density, for example if fish passage through a fishway or culvert is impeded by having too many fish trying to access it at once. By “density” we meant a neutral measure of the number of fish (per unit space) present. To make this clearer we have changed all instances of “crowding” to “overcrowding”. Further, we make this negative connotation for “overcrowding” explicit in e.g. line 316.

Please write out specialized abbreviations at their first use like DART (data access in real time),

Addressed for DART, USGS, NWIS at lines 123–125. Acronym GLM removed. Addressed for PH at line 162. Addressed for dams at line 114.

As mentioned before, I think using the combination of two different types of measurements is a great approach in this manuscript, but the differences between what DART and telemetry provide (advantages and drawbacks) should be explained in a bit more detail (especially for non-specialist readers), and therefore what you gain by using both.

We have added lines 125–134, highlighting the different roles that counts and telemetry play, as well as foreshadowing the challenges that arise from such a synthesis.

Could there be learning effects between dams? Can you comment on this?

The role of learning in the context of social behavior and collective movement is an on-going area of research (e.g. Sasaki & Biro. 2017. *Nature communications* 8 (1), 1-6.), but for a variety of reasons we do not think learning is a major contributor to the patterns revealed in our manuscript. First and foremost, the dams on the mainstem Columbia are very different from each other, with each fishway and tailrace being unique. Thus the challenge of locating and entering the fishways differs among dams and the knowledge gained by ascending previous dams is unlikely to greatly aid navigation over subsequent dams. Second, the antenna configurations and locations of fishway openings at the dams probably drives the differences in fish passage times between dams more than anything, especially the distance the tailrace antennas were located downstream from the dams, which makes taking into account the potential role of learning dubious at best.

As a consequence of all above, I suggest acceptance after minor revision. As the changes suggested are minor, the Editor may decide to accept them without my further involvement. But I am also happy to check the revised version of this manuscript.

3 Response to Referee 2

Review of the manuscript titled Collective navigation facilitates passage through human-made barriers by homeward migrating Pacific salmon.

General comments: I think the topic of collective navigation for migrating animal species is of great interest, especially whether travelling in groups influences the ability of these species to pass challenging areas during their migration journey. The current study aims at answering this question for two species of Pacific salmon migrating in freshwater and encountering hydropower dams. While I think the topic is very important, I feel this work should include other species or other systems in order to be considered for publication in ProcB. Perhaps passage by other species was also monitored by radio telemetry and daily counts at viewing windows for the Columbia River dams? Or at dams in other river systems in North America or Europe? If so, analyzing more data with the proposed approach could allow the authors to uncover positive effects of density on passage performance for a range of species, or even species-specific response to density with regards to passage at dams. This would lead to a very strong paper with a broader perspective. I understand that what I suggest means conducting a new series of analysis and possibly collaborating with other researchers in the case of including data from other systems.

We concur that it would be of great value to analyze data from other species and other systems using our approach, and have added reference to such to our text at lines 335–336. Our contribution is as much methodological as it is scientific and we hope our methods will be of use to others in the future. We decided it was not feasible to add additional species or systems for the following reasons.

Firstly the telemetry dataset we worked with is quite unique, and few comparable data sets in other systems exist to synthesize with our own. What data sets may exist are likely different in many ways that would obfuscate statistical analysis. We have pieced together data from several different sources to conduct our analysis, but each covariate was pulled consistently from a single source, which lends confidence that each covariate has a consistent and stable interpretation. This is lost once we begin to include data from other existing projects. Follow up work in new systems could involve collection of new telemetry data, but that was certainly outside the scope of this project.

Staying within the telemetry data set that we analyzed, we could have included data from other species, however these posed further problems. In some cases, other species pose quite different questions: for example, steelhead are partially iteroparous, and this might contribute to a confounding learning or leadership effect. Other species, such as American shad, were telemetry tagged only on an ad hoc basis, and small sample sizes would plague our analysis, which relies on large sample sizes to fit complex models. Furthermore, we have the most experience with salmon biology and behaviour and would have a much more difficult time interpreting results from shad or other such species.

For these reasons, and taking into account the editor's expressed leniency on this point, we

have not updated our analysis in this manner.

I am thus suggesting a major revision/reframing of the manuscript, or a transfer to another journal if more data cannot be included. Below are some specific comments that may inform the revision.

Specific comments:

Manuscript structure: I find the manuscript unbalanced, with a reasonable introduction, a long method section, and quite short results and discussion sections. I would have liked to see more figures in the results, in particular of the survival analysis models, and a more developed discussion.

See our updated results and discussion sections. The discussion section in particular we have expanded substantially. We found that the results section was necessarily brief compared to the methods section — use rather complicated methods to arrive at a rather simple numerical result — and found that attempts to lengthen it resulted in material more suitable to the methods or discussion. We agree that the methods section is lengthy but found it difficult to reduce the size while also addressing parallel comments regarding accessibility to a broad audience (see e.g. lines 146–174 where we introduce the basics of time-to-event modeling) and justification of our methods (see e.g. lines 207–218 where we justify our bootstrap approach). To address this point we have moved most of the more technical model building discussion to ESM, leaving only the bare minimum in section Statistical Analysis and including section Modeling Approach to explain the conceptually underpinnings of our models.

Abstract and Introduction

1.23: other major drivers would be flow velocity and diel effects for some species

This is true. We do not mention diel effects here because in our models they were much stronger than density effects. We do not mention flow velocity because we did not include this as a covariate. However see updated line 369 where we acknowledge that other covariates, including flow velocity, may have been important.

1.38: river passage? Perhaps a term such as upriver migration would be more adequate.

We have changed our wording on line 42 to more precisely note the effect which dams have on both upriver and downriver migration.

1.42-43: I like that these mechanisms are mentioned here. However, I think adding 2 sentences to briefly define some (ex: the many wrongs principle) would be worth it.

We have added a brief description of the ‘many-wrongs principle’ and of ‘emergent sensing’, which we believe are the two most relevant mechanisms for the current situation. Further we

refer interested readers to a detailed description of all of the mechanisms in a recent review of the topic (lines 46–53).

1.62: perhaps extend to population conservation?

We have included reference to population conservation at lines 71–72.

1.66; fish ladders...I would generalize to hydropower complexes in regulated rivers

We feel that our study is focused specifically on pool-and-weir fishways (fish ladders) within hydropower complexes. Although they are relatively common, not all hydropower complexes have fish ladders, and not all mechanisms for fish passage past hydropower complexes are fish ladders. We have therefore chosen not to generalize in this section.

Method

Figure 1. This figure is nice. However: - it is missing some kind of a scale, or at least a measure of distance for the whole hydropower complex (from the beginning of the tailrace to the upstream reservoir where there is no human structure). - the distinction between fishway and fish ladder is not well illustrated. Does the fishway start before the ladder? This is unclear on the figure. - Also, is it possible to include your telemetry array on the figure?

We have added to the figure caption a measure of distance from the tailrace antenna to the dam. We have also added to the figure the approximate locations of our telemetry array. We have clarified our use of “fishway” and “fish ladder” elsewhere in the manuscript, and we are grateful for this feedback on that point. The fishway does indeed start before the fish ladder, including also the entrances, collection channels, and junction pools. We had previously carelessly used the two almost synonymously, and have corrected this usage in our resubmission. Rather than distinguish the ladder proper from the remainder of the fishway in the figure, we have attempted to clarify in the remainder of the text that in fact the entire fishway passage process is what we are modeling in our “fishway” system (previously, “ladder” system).

1.94: Here you mentioned a travel speed in undammed area of the river. It would be useful to know the length of a whole dam complex including tailrace and near-dam reservoir (5 -7 km?) to make a fair comparison. The difference in travel speed (or transit time) is high (approx. 2 km/day vs 18-54 km/d)

We have added the average length of the Columbia/Snake River dam complexes (0.5–3.2 km) and the associated speeds during dam passage (0.17–3.2 km/day) to make it more explicit that the dams result in a 10–100-fold reduction in speed (lines 101–107).

1.99 during their upstream migration

See updated line 112.

1.112-118: have you considered using the term time-to-event analysis framework instead of

survival? The analysis framework comes from the medical literature and was traditionally used to measure survival of patients with different treatments. However, it has been also used in social studies and other fields to measure the rate of occurrence of various events involving a time component (transition from one state to another).

See lines 146–174 where we have transitioned mostly to using the time-to-event analysis.

If I understood well, your dam system is divided into multiples systems; tailrace, fishway & upstream reservoir and you analyzed the rates of transition between each of those, specifically fishway entry rate, and fishway/fish ladder passage. You also quantified how time-varying covariates (temperature, spillway, fish density, diel period) and some static covariates (species, dam, run timing). No random effects were added to look at unexplained variability. A logistic model was also used to look at fish commitment to pass the fishway in one attempt. Given the reverse causality issues, a bootstrap approach was then used to detect density effects. I have a few questions/concerns:

- Why not also measuring passage (transit) through the tailrace?

We were limited by the precision of our telemetry array, and unfortunately had no clear way of judging when a fish had transited the tailrace and begun searching for the fishway entrance. Future studies might use finer-scale data to distinguish these processes, or incorporate a more sophisticated model which might include both separate processes as separate states in a multi-state model with unobserved state transition times, but this was beyond the scope of our analysis here.

- What was the measuring interval for counts? Were counts merged with the movement database and treated as a time-varying covariate? L.65 mentioned daily counts but I feel it was measured more often.

Yes, counts were merged with the movement database and treated as a time-varying covariate. You are corrected that counts are also reported at a finer (hourly) temporal scale. However, see lines 198–206. We did not use these finer scale counts because their relationship with density is less consistent and the time-delay effect is more pronounced.

- Why a binary diel effect? From my experience, fish often move at transitional periods (dawn and dusk) and a 3 or 4 levels variable is often more efficient.

Previous work on this study system (Keefer et. al, 2013a) indicates that although crepuscular activity does occur, the binary effect appears to be strongest and most parsimonious for the processes we are interested in. Informal plots of our data confirm that this binary diel effect seems to match our system most closely. See lines 200–201 where we address this to some degree. An overwhelming majority of fish pass during the daytime, and crepuscular patterns did not appear to explain this.

- Why not treating species as a fixed effect in your models?

See updated ESM line 10. We expected that species would differ not only through some fixed difference in rate, but also through different interactions with environmental covariates. In principle we could have treated species as a fixed effect, but based on this expectation we would have had to include interactions between species and each covariate. In practice it was simpler to just have separate models for each species.

- It would be good to add a schematic of your analysis as a main or supplemental figure. You ended up splitting your dataset into multiples ones and fitting separate models to those. While I do not question this decision, it makes things a bit confusing for the reader.

We are grateful for this feedback as we had struggled to present our results in a clear way, in part as a result of poor communication of which models we ultimately used. We have attempted to provide a clearer accounting of our model splitting procedure in Figure 1 in our ESM.

- Did you conduct some kind of model selection on your proportional hazards and logistic models in order to select the best-fitting one for each dataset? If so, these results should be presented.

We did not previously use model selection because most of our covariates (`temperature`, `day`, `spill`) have been included in a number of previous models and are established as having some influence in these processes. Furthermore, model selection poses some slight difficulties as we wish to select variable sets that are as consistent as possible when models have been split. However, upon further reflection, we decided to improve our analysis by using model selection for our null models, and to select our density variable (`count` vs. `log-count`). See ESM lines 28–36. Previously we used raw count in all models, and this did not change when we considered using `log-count`. After model selection, we dropped `middle` from most of our models, as well as the `Okanogan/Wenatchee/Other` factor variable.

- When using the parametric bootstrap, you mentioned null models and full models. Is a null model the selected PH model without fish density, or the PH model with all possible covariates except fish density? Clarification is needed here.

We have added clarification for the meaning of null model vs. full model in lines 218–220. A null model is any model without fish density and we have tried to specify the “selected” null model when necessary.

- I like seeing how you dealt with reverse causality in your system. However, I am not very familiar with the parametric bootstrap approach and I feel it should be defined more to help the reader understand the results.

We clarify the parametric bootstrap approach in lines 207–218.

- L.190: you mentioned applying threshold for the commit models for Chinook, due to substantially higher commit rates for these low-density fish I am unsure what this means...

We have addressed the threshold effect in more detail in lines 351–361. We agree it is deserving of more attention than was initially afforded it. We spent little time discussing it initially because, for reasons we state in the text, we feel there is not enough evidence to draw conclusions about the true presence of a density threshold effect. We feel that this is an intriguing avenue for further study, but treat it in our analysis largely as just another confounding variable for inclusion in the null model.

Results

- The results from the parametric bootstrap tests are almost exclusively presented, along with two sentences about the effect of other covariates in the PH models. I was hoping for more...especially I was hoping to see some entry rate, or passage rate curves adjusted for the effect of covariates. For example, a figure showing entry or passage rates over time (proportion of your available population entering or passing) for 3 different fish densities (while controlling for the other covariates effects) would help the reader visualize the density effect. This figure could also show both species, allowing the reader to visualize the species-specific response to density. - Overall, I would suggest to present the bootstrap results in a multi-panel figure, then some time- to-event response curves adjusted for the effect of covariates in another multi-panels figure.

We are particularly grateful for this feedback, as we have recognized the need for a more representative figure but have struggled to produce it. We thank you for providing the last necessary push to motivate us to work out how to do this. We were able to produce Figure 3 by making use of the same parametric bootstrap we used for other portions of the manuscript, simulating from our fitted model for each individual fish, thereby averaging over the effect of other covariates. We initially produced these figures for varying levels of constant fish densities, but ultimately decided to go with varying instead a density *factor*. This allows the shape of the density curves over time to remain constant, as well as the relative timings between highs and lows of density with respect to other seasonally-varying variables. We believe this provides a more accurate representation of what our model predicts with varying run sizes.

Discussion

The discussion is not enough developed in my opinion: - Pacific salmon are migrating upstream in large numbers, which is similar to other fish (American shad, lampreys, herrings, etc.) or animal species (geese, butterflies), but different from others who can travel long stretches alone (Atlantic salmon, sturgeon, anadromous trout). How do you think collective navigation would be an advantage for them, especially at barriers that may delay the overall rates of movements?

We have expanded our Discussion dramatically, including additional discussion of what mechanisms might underlie the density effects we observed. We are, however, uncomfortable speculating too broadly about how collective navigation might benefit other fish or animal

species, since our analysis only includes Pacific salmon. We have drawn comparisons to some fish species, and are content to suggest that our results should motivate further work on other species.

- How can your results affect management and design of fish passage structures: if collective navigation is a positive asset for Pacific salmon, how can managers include this in fish passages design to promote higher entry and passage rates?

We are grateful for this feedback, as it was an appropriate reminder to us not to overreach when describing the impacts of our research. We have removed all references to design of fish passage structures, as we are indeed unsure of the specifics of how our results might inform such designs. We have addressed potential use of these results by managers in two parts: lines 345–350 and lines 382–391. On the one hand our results inform more accurate predictions of the energetics and ultimate success of salmon migration. In lines 382–391 we address how critical a density effect may be in such predictions, since they can yield threshold effects which would not otherwise occur. On the other hand, further research into the *mechanisms* underlying these density effects may inspire management interventions to improve salmon migratory success. We discuss one hypothetical intervention in lines 345–350.

Appendix B

Response to Reviewers

Connie Okasaki, Matthew Keefer, Peter Westley, Andrew Berdahl

September 29, 2020

We would like to thank the editor and referees again for a productive and prompt review process. Below, please find our (very brief) responses to the proposed minor edits. Unrelated to the proposed revisions, we also cut approximately 400 words in order to satisfy the strict 10-page limit which we are just on the edge of.

1 Response to the Editor

The revised manuscript is much improved. There remain a few minor suggestions from the referee. I also recommend the final sentence of the abstract needs to be changed from "...in at least some species,..." to "...in at least one species,..." as the positive results are for Chinook salmon only.

We have changed this sentence as suggested, at line 31.

2 Response to Referee 2

1.124 : Briefly mention which environmental data.

We have added that the environmental data is specifically temperature and spillway discharge at new line 124.

1. 143: remove ‘also known as survival’ as you repeat it a few lines below.

We have removed this as suggested, at line 142

1.155: ‘find and enter a fishway’? Also, what is the question of your second model?

We have updated our language at line ???. We have also added the question for the second model to this section.

1.158: there is a missing word in this sentence (we?)

We fixed this typo at line 154.

1.222: The ESM is well made. However, I would still mention how you splitted the data (fishway ID, run timing) as it will help the readers understand Table 1.

We have added lines 206–211 to give additional detail about how we split our model as well as what we mean by “splitting” a model (really it is splitting the dataset for that model into disjoint subsets).

1.224: when you say ‘test’, do you mean model or testing the effect of density? In other words, do ‘test’ and ‘model’ mean the same thing here?

Thanks for catching this. “Test” and “model” do mean different things here. We ran one null model test on each model, so the number of them is the same. We have updated the text to clarify this at lines 212–213.

1.241-245: I find this paragraph a little bit difficult to read. For example, when you say ‘test’, do you mean the ‘model’? You mention ‘highly significant test’. Do you mean the density effects were significant for these tests? What are the two other significant effects on line 243? I think some clarifications are needed.

We have updated this paragraph at lines 228–235. We removed “final” to avoid implicitly equating our tests with our models (though they are closely related). We have changed our verbage about test significance, and specified the two other effects that we mention.

1.245: density effect are positive, meaning that an increase in density will increase the rate of finding the fishway, or the probability to commit to pass the fishway etc. Sometimes, it

helps the readers to define the direction of a coefficient, even if it seems obvious to us.

Indeed! We have added this clarification at lines 234–235. We agree this will be helpful for readers.

1.265-271: Do you provide the coefficients for the other covariables (diel, water temperature, etc.) somewhere? Perhaps it could go in supplemental material.

We have added our final best-fitting model summaries to the end of the ESM.

1.265: do you mean previous studies in the same system?

Yes, we have updated to reflect this at line 254.

1.267. You mention a strong diel effect. Which direction? More passage during the day?

We have updated the text at line 256 to reflect that we mean higher passage during the day.

Table 1: sample size is the number of fish radio-tagged. Maybe mention it in the caption.

We have updated the caption with this clarification.

1.322: ‘zero density effect’. You mean no effect. . .

Yes this is more accurate and we have changed this at line 307.

1.325: actually, fish were confined (caged) in both studies.

We’ve updated the parenthetical at line 309 to reflect this.

1.351-361: you mention an ‘additional effect’. Are you talking about the threshold for Chinook salmon? I think you should describe the use of this threshold better, both in methods and in discussion. Without more information, this paragraph is hard to understand. The threshold use is explained in ESM, but not all readers will consult this document.

We have tried to specify more clearly, particularly in lines 335–343 when we are discussing the threshold effect. However, although perhaps more technical in parts, we feel that the threshold information is all present in either the methods or discussion. Little additional information is presented in ESM.

1.371: I am unsure about the use of ‘recreation’. Maybe ‘representation’? This is only a suggestion.

We agree this is a better word choice and have changed line 352 accordingly.